# Seroprevalence of Dengue, Chikungunya and Zika at the epicenter of the congenital microcephaly epidemic in Northeast Brazil: A population-based survey

Cynthia Braga[1]*, Celina M. T. Martelli[2], Wayner V. Souza[2], Carlos F. Luna[2], Maria de Fatima P. M. Albuquerque[2], Carolline A. Mariz[1], Clarice N. L. Morais[3], Carlos A. A. Brito[4], Carlos Frederico C. A. Melo[5], Roberto D. Lins[3], Jan Felix Drexler[6,7], Thomas Jaenisch[8,9,10], Ernesto T. A. Marques[3,11]*, Isabelle F. T. Viana[3]

1 Department of Parasitology, Institute Aggeu Magalhães, Oswaldo Cruz Foundation, Recife, Pernambuco, Brazil, 2 Department of Public Health, Institute Aggeu Magalhães, Oswaldo Cruz Foundation, Recife, Pernambuco, Brazil, 3 Department of Virology, Institute Aggeu Magalhães, Oswaldo Cruz Foundation, Recife, Pernambuco, Brazil, 4 Department of Clinical Medicine, Federal University of Pernambuco, Recife, Pernambuco, Brazil, 5 Pan American Health Organization, Brasília, Federal District, Brazil, 6 Charité—Universitätsmedizin Berlin, Corporate Member of Freie Universität Berlin and Humboldt-Universität zu Berlin, Institute of Virology, Berlin, Germany, 7 German Centre for Infection Research (DZIF), associated partner site Charité, Berlin, Germany, 8 Section Clinical Tropical Medicine, Department of Infectious Diseases, Heidelberg University Hospital, Germany, 9 German Centre for Infection Research (DZIF), Heidelberg Site, Heidelberg, Germany, 10 Center for Global Health, Colorado School of Public Health, Aurora, Colorado, United States of America, 11 Department of Infectious Diseases and Microbiology, University of Pittsburgh, Pittsburgh, Pennsylvania, United States of America

* cynthia.braga@fiocruz.br (CB); marques@pitt.edu (ETAM)

**Data Availability Statement:** All relevant data are within the manuscript and in the Supporting Information files.

## Abstract

### Background

The four Dengue viruses (DENV) serotypes were re-introduced in Brazil's Northeast region in a couple of decades, between 1980's and 2010's, where the DENV1 was the first detected serotype and DENV4 the latest. Zika (ZIKV) and Chikungunya (CHIKV) viruses were introduced in Recife around 2014 and led to large outbreaks in 2015 and 2016, respectively. However, the true extent of the ZIKV and CHIKV outbreaks, as well as the risk factors associated with exposure to these viruses remain vague.

### Methods

We conducted a stratified multistage household serosurvey among residents aged between 5 and 65 years in the city of Recife, Northeast Brazil, from August 2018 to February 2019. The city neighborhoods were stratified and divided into high, intermediate, and low socio-economic strata (SES). Previous ZIKV, DENV and CHIKV infections were detected by IgG-based enzyme linked immunosorbent assays (ELISA). Recent ZIKV and CHIKV infections were assessed through IgG3 and IgM ELISA, respectively. Design-adjusted seroprevalence were estimated by age group, sex, and SES. The ZIKV seroprevalence was adjusted to account for the cross-reactivity with dengue. Individual and household-related risk factors

**Funding:** This study was funded by European Union's Horizon 2020 Research and Innovation Programme under the ZIKAlliance Consortium (grant number H2020 734548 to T.J. and E.T.A. M.); the German Centre for Infection Research (DZIF, Heidelberg Site, Germany, to T.J. and J.F. D.); the Pan American Health Organization, World Health Organization/Brazilian Ministry of Health (grant number SCON2018-00276 to C.B.); the National Council for Scientific and Technological Development (Conselho Nacional de Desenvolvimento Científico e Tecnológico – CNPq; grant numbers 303953/2018-7 to C.B.; 308000/2021-8 to W.V.S.; 302696/2021-0 to M.F.P.M.A.; 303001/2018-6 to R.D.L.; 425997/2018-9 to R.D.L. and INCT-FCx to R.D.L.); the State of Pernambuco Funding Agency (Fundação de Amparo à Ciência e Tecnologia do Estado de Pernambuco - FACEPE; grant number BFP-0010-2.11/22 to I.F.T.V. and APQ-0346-2.09/19 to R.D.L.); the Oswaldo Cruz Foundation through the Innovation Program (INOVA, grant numbers VPPCB-007-FIO-18-2-134 to R.D.L. and I.F.T.V.; IAM-005-FIO-22-2-44 to R. D.L.). The content of the present study is solely the responsibility of the authors and does not necessarily represent the official views of the funding agencies. The funders had no role in study design, data collection and analysis, decision to publish, or preparation of the manuscript.

**Competing interests:** The authors have declared that no competing interests exist.

were analyzed through regression models to calculate the force of infection. Odds Ratio (OR) were estimated as measure of effect.

## Principal findings

A total of 2,070 residents' samples were collected and analyzed. The force of viral infection for high SES were lower as compared to low and intermediate SES. DENV seroprevalence was 88.7% (CI95%:87.0–90.4), and ranged from 81.2% (CI95%:76.9–85.6) in the high SES to 90.7% (CI95%:88.3–93.2) in the low SES. The overall adjusted ZIKV seroprevalence was 34.6% (CI95%:20.0–50.9), and ranged from 47.4% (CI95%:31.8–61.5) in the low SES to 23.4% (CI95%:12.2–33.8) in the high SES. The overall CHIKV seroprevalence was 35.7% (CI95%:32.6–38.9), and ranged from 38.6% (CI95%:33.6–43.6) in the low SES to 22.3% (CI95%:15.8–28.8) in the high SES. Surprisingly, ZIKV seroprevalence rapidly increased with age in the low and intermediate SES, while exhibited only a small increase with age in high SES. CHIKV seroprevalence according to age was stable in all SES. The prevalence of serological markers of ZIKV and CHIKV recent infections were 1.5% (CI95%:0.1–3.7) and 3.5% (CI95%:2.7–4.2), respectively.

## Conclusions

Our results confirmed continued DENV transmission and intense ZIKV and CHIKV transmission during the 2015/2016 epidemics followed by ongoing low-level transmission. The study also highlights that a significant proportion of the population is still susceptible to be infected by ZIKV and CHIKV. The reasons underlying a ceasing of the ZIKV epidemic in 2017/18 and the impact of antibody decay in susceptibility to future DENV and ZIKV infections may be related to the interplay between disease transmission mechanism and actual exposure in the different SES.

## Author summary

The extent and population burden of the Zika and Chikungunya epidemics in Northeast Brazil remains speculative since seroprevalence studies have often been restricted to specific populations and limited by ZIKV and DENV antibody cross-reactivity. Here we conducted a seroepidemiological study in the city of Recife, a metropolitan area in Northeast Brazil using a design stratified by socioeconomic status (SES). We determined the sensitivity and specificity of the assays using a panel of well-characterized samples from the study area and determined optimum cut-offs, which were later validated by selecting a subset of samples that were used to conduct viral neutralizations assays. The results indicated that 89% of the population (older than 5 years of age) had previous dengue infection, which is compatible with our previous serosurvey. Correcting for the sensitivity and specificity of the ZIKV assays, the overall ZIKV IgG seroprevalence was 34.6%, which indicates high transmission during the first outbreak (2015/2016). Interestingly, the age and the SES distribution profiles of ZIKV and CHIKV seroprevalence were remarkably different. This difference cannot be explained by differences in mosquito exposure alone. Future research will need to be conducted to better explain the differences found in the age distributions.

## Introduction

Arthropod-borne diseases, especially dengue, chikungunya and Zika, have presented a major public health problem in the Americas [1]. Brazil has been responsible for more than 90% of the reported cases of these arboviruses in this region in the last decades [2].

The recent introduction of Zika virus (ZIKV, genus *Flavivirus*, family *Flaviviridae*) in the Northeast Region of Brazil, around 2013 and 2014, was followed by its rapid spread to other regions causing significant increase in the number of cases of Guillain-Barré syndrome in adults and congenital microcephaly secondary to maternal ZIKV infection during pregnancy [3,4]. The Zika outbreak was followed by an outbreak of the Chikungunya virus in Recife (CHIKV, genus *Alphavirus*, family *Togaviridae*) in 2016 that suppressed ZIKV transmission [5]. Data from the Brazilian Ministry of Health show the states of Bahia, Ceará and Pernambuco, all in the Northeast region, as the most affected by the ZIKV and CHIKV epidemic in Brazil [6,7].

Population-based seroprevalence surveys stratified by age and geographic areas have been considered one of the current research priorities, due to their ability to estimate variations in the level of exposure of the population according to time, age, and environment [8,9]. These studies provide much anticipated granular data that will allow to infer more precisely the susceptibility and level of immunity in each population group [8].

In Brazil, the true magnitude of the ZIKV and CHIKV outbreaks and the extent of the ensuing low-level transmission is still not well known [10]. The city of Recife, a large urban center marked by profound social inequalities in the Northeast region of Brazil, has been affected by successive arbovirus epidemics since the introduction of the Dengue virus (DENV) in the 1980s [11]. A dengue serosurvey conducted between 2005 and 2006 in three socioeconomically distinct neighborhoods of Recife estimated an overall prevalence above 80% [12] and showed an inverse association between the force of infection and socioeconomic status (SES). Between 2015 and 2016, with the emergence of ZIKV and CHIKV in Brazil, more than 50,000 cases of dengue, Zika and chikungunya were registered in this city [13,14]. Those also accounted for around 90% of the cases of microcephaly attributed to ZIKV in Brazil [15,16], of which 97% occurred in babies of mothers with low SES. The reason for this disproportional distribution remains unclear, and one hypothesis is that pregnant women of high SES were much less exposed to Zika. We now carried out a detailed population-based survey in the whole city to determine the levels of exposure to ZIKV, DENV and CHIKV according to age and SES using well characterized serological tests. The resulting data was analyzed according to the sensitivity and specificity of each assay.

## Methods

### Ethical statement

This research was approved by the Research Ethics Committee of the Aggeu Magalhães Institute (IAM, Fiocruz, Pernambuco) (CAEE: 79605717.9.0000.5190, report number 2.734.481). Data collection was conducted only after the participants or their legal guardians (if under 18 years old) were informed about the objectives of the study, read and sign the consent form. Participants aged 5 to 18 years provided oral and/or written assent. All participants had access to the results of the laboratory tests. Personal information was removed prior to data analysis.

### Study design, population, and settings

The seroprevalence survey was conducted using a stratified multistage cluster sampling design involving residents aged 5 to 65 years old, from August 2018 to February 2019. Recife, the

capital city of the Pernambuco state, has a territorial area of 218.8 km$^2$ (divided in 94 neighborhoods), an estimated population of approximately 1.6 million inhabitants and a demographic density of 7,037.6 inhabitants/km$^2$. The city is classified as the 12$^{th}$ most densely populated urban area in Brazil. Around 40% of its population lives in poverty, with a monthly income of up to half minimum wage, while 30% of households are in areas with inadequate sanitation [17].

## Sampling

The city's neighborhoods were stratified into three economic strata based on the information of the family's head income per census tract obtained in the 2010 Demographic Census. The census tract (CT) is the smallest territorial unit for obtaining population data and has approximately 300 households (or approximately 1,000 inhabitants) [18].

Briefly, to split the city territory at neighborhood level, the percentage of households whose head of family had no income or declared to have a monthly income <2 minimum wage (MW) per CT was first calculated. Subsequently, these CTs were aggregated at neighborhood level and classified into four clusters relatively homogeneous regarding socioeconomic status, using k-means clustering technique and ANOVA test [19]. Further details on the methodology to stratify the city's territory were described elsewhere [20]. In this study, the high socioeconomic stratum was composed by the two clusters of neighborhoods with the highest household income (Clusters 1 and 2), due to the small population size in both individual clusters. The 3$^{rd}$ and 4$^{th}$ clusters were classified as intermediate and low socioeconomic strata, respectively.

The population sample size in each stratum was calculated considering an expected seroprevalence of 30% in the high socioeconomic stratum, and of 40% in the intermediate and low socioeconomic strata; absolute error of 4%; design effect of 1.5 and 95% confidence level. The final sample was comprised of 2,500 participants including 760 individuals in the high socioeconomic stratum, and 870 individuals in both intermediate and low strata.

The number of CTs (which represent the primary sampling units) to be selected in each stratum was determined considering an estimated prevalence of arbovirus infected individuals of 60% in the high socioeconomic stratum, and of 80% in both intermediate and low socioeconomic strata per CT. The final sample was comprised of 40 CTs in the high socioeconomic stratum and 30 CTs in the intermediate and low strata, therefore representing a total of 100 CTs across the city.

The number of households to be selected in each CT was determined based on the estimate of 3.5 inhabitants per eligible household. The final sample was comprised of 716 households, including 218 households in the high socioeconomic stratum and 249 households in the intermediate and low strata. The participants selection was conducted in two stages, where we firstly randomly selected the CT in the strata (first stage) and then (second stage), we selected the households (including all residents in the study age group) within the CT using the Amostra Brasil free R [21] package for household sampling (with their respective geographic coordinates) in Brazilian municipalities from the IBGE database [22]. Fig 1 shows the spatial distribution of the sampled households according to the socioeconomic strata in the city.

## Data collection

The field team, comprised of interviewers and phlebotomists, was trained to ensure standardization and data quality. Home visits were initially performed to inform the head of the family about the objectives of the study and to invite the eligible residents to participate. Individual and household information were collected using standardized questionnaires covering a list of exposure variables relevant to the study (S1 Table). After the interview, a venous blood sample of the participants (8 mL of adults and 5 mL of the children aged up to 9 years) was collected

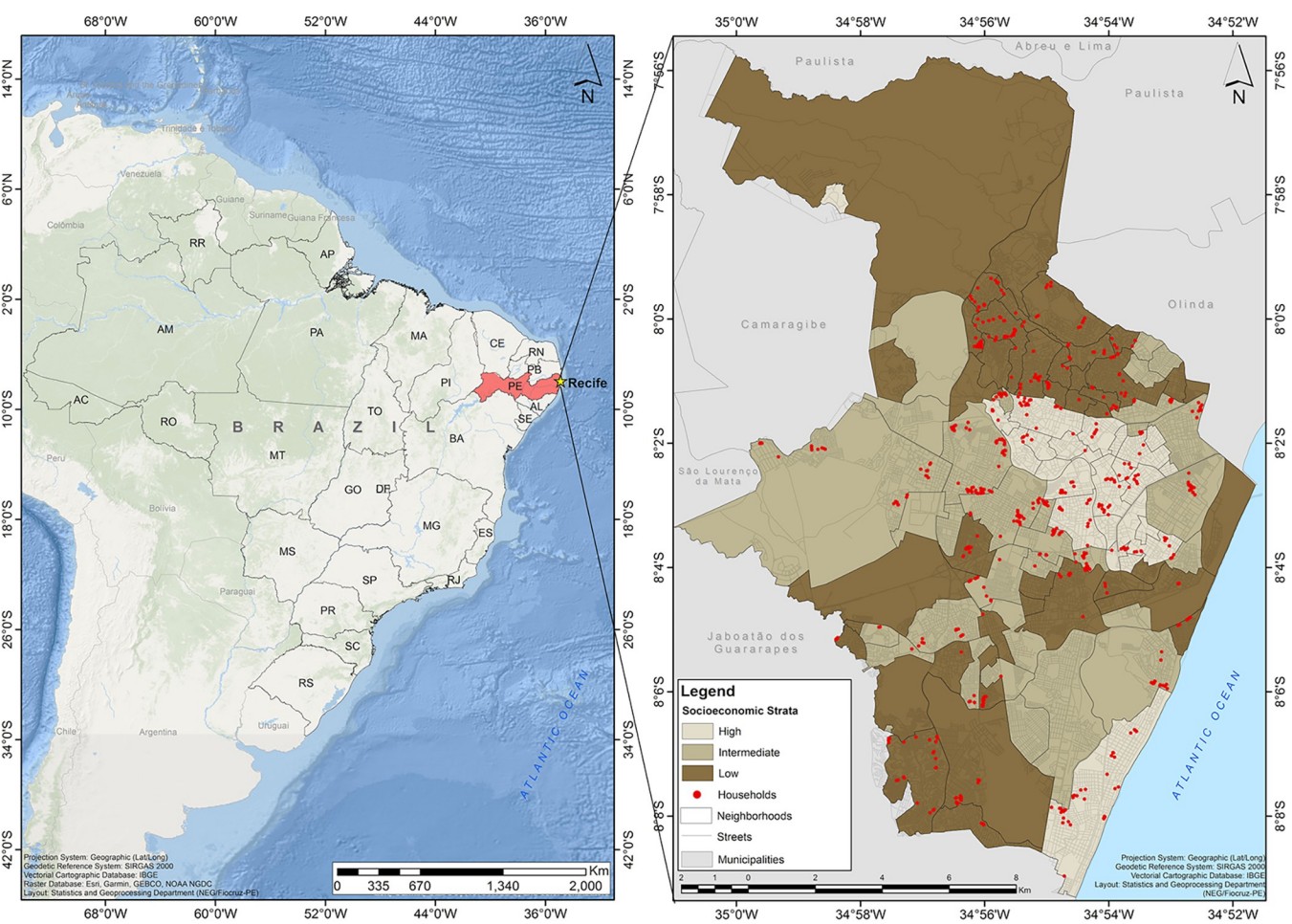

**Fig 1. Spatial distribution of the selected households according to socioeconomic Strata, Recife, Brazil.** The geodetic reference system SIRGAS2000 (Geocentric Reference System for the Americas) was the coordinated system used to represent geometric or physical terrestrial characteristics (http://www.ibge. gov.br/english/geociencias/geodesia/pmrg/faq.shtm#1).

to perform the serological tests. Samples were collected in vacuum tubes with clot activator for 30 minutes at room temperature and then placed in a container at 4˚C. Urine and hair samples were also collected from the participants that showed signs of recent infection and transported to the Department of Virology at the Aggeu Magalhães Institute, Fiocruz, where they were processed and stored at -70˚C until use.

## Laboratorial procedures

Anti-CHIKV IgG and IgM antibodies were detected through commercial ELISA kits (Euroimmun, Lubeck, Germany). The results of both tests were interpreted according to the manufacturer instructions (CHIKV IgG or IgM absorbance at 450 nm/calibrator ratios were considered negative at <0.8, undetermined at ≥0.8 to <1.1, and positive at ≥1.1). All samples with indetermined results were retested using the same commercial kits, and the obtained results were considered as final.

The anti-ZIKV immune response was assessed by detecting IgG through a commercial ELISA kit (Euroimmun, Lubeck, Germany) [23]. Aiming to overcome the possible cross-reactivity between anti-DENV and anti-ZIKV immune responses, the cut-off of this test was

redefined using a panel of 140 well-characterized serum samples from acute and convalescent samples of PCR confirmed DENV infection cases collected years prior to the Zika outbreak and samples from PCR confirmed cases of ZIKV infections. Results were interpreted as positive for ZIKV IgG when the sample absorbance at 450nm/calibrator ratio $\geq$1.35. Based on this cut-off point, the estimated sensitivity and specificity of this test were 86% (95%CI: 72–95%) and 72% (95%CI: 65–79%), respectively (S1 Fig).

Recent ZIKV infections were determined by detecting IgG3 antibodies against the ZIKV NS1 protein through an in-house ELISA [24–26] (S1 Additional methods). Results were interpreted as positive for ZIKV IgG3 when the sample absorbance at 450nm/DENV recent infection control ratio $\geq$1.14. According to this cut-off, the estimated sensitivity and specificity of this test were 81% (95CI%: 60%-95%) and 93% (95%CI: 88%-96%), respectively (S1 Fig). The choice of this test over the commercially available ZIKV IgM kit (Euroimmun, Lubeck, Germany) was due to the lower sensitivity of the latter in our analysis using a well-characterized serum panel, where the lower sensitivity of this test in a scenario of flavivirus co-circulation has been documented [27,28].

The accuracy of the ZIKV serology results was validated by performing a blind plaque reduction neutralization test (PRNT) of a subset of 156 randomly selected serum samples and compared with the other serological tests. The PRNT was performed following a modified protocol described in detail elsewhere [29] and neutralization was assessed against the ZIKV local strain (BR-PE243/2015). The cut-off for PRNT positivity was defined based on a 50% reduction in plaque counts (PRNT50), and ZIKV-specific antibody titers were estimated using a four-parameter non-linear regression. Samples were considered positive when the PRNT50s were $\geq$1:100 for ZIKV (S1 Additional methods) corroborating sensitivity and specificity data. The duration of ZIKV binding and neutralization antibodies two years after infection have been also evaluated in a previous study that showed significant decay of antibody levels but very few seroreversions [30].

Previous exposure to DENV was assessed by detecting IgG against the DENV 1–4 NS1 proteins through an in-house indirect ELISA, as described elsewhere [31]. Samples were considered positive for DENV 1–4 IgG when sample absorbance at 450nm/positive control ratio $\geq$3.62, corresponding to a sensitivity of 96% (95% CI:90–99%) and specificity of 71% (95% CI: 58–82%) (S1 Fig).

## Laboratorial classification of samples

Serological and/or neutralization positive samples only for ZIKV or only for CHIKV were classified as ZIKV+ and CHIKV+, respectively. Negative samples for ZIKV and CHIKV tests and positive for DENV 1–4 ELISA test were classified as ZIKV-/CHIKV-/DENV+. Samples that were negative for all arbovirus tested were classified as ZIKV-/CHIKV-/DENV-.

## Exposure variables

We considered the following set of exposure variables at household and individual levels: i. household level: residents per bedroom, type of household, sewerage destination, water supply, frequency of water supply, garbage collection, sociodemographic characteristics of the head of the family; ii. individual level: age (in years), age group (5–14, 15–24, 25–34, 35–44, 45–54, 55–65), gender, self-reported skin color, schooling ($\geq$13 years old), previous dengue infection, use of repellent, previous dengue exposure, vaccination for yellow fever virus or DENV. Participants who reported fever or skin rash in the 30 days prior the interview were asked about other clinical manifestations that may indicate arbovirus infection (dengue, chikungunya or Zika) and were requested to provide urine and hair samples (S1 Table).

## Statistical analysis

Double data entry and data consistency analysis were performed using REDCap electronic data capture tools [32] hosted at Heidelberg University, Germany. The analysis was performed using the R software, version 4.0.27 [21].

## Seroprevalence estimates

The seroprevalence of dengue (IgG), Zika (IgG and/or IgG3) and chikungunya (IgG and/or IgM), and their respective 95% confidence intervals (95%CI), were estimated according to age group and sex for each socioeconomic stratum (high, intermediate and low). These estimates were weighted by the effect of the sample design using the "survey" package, version 4.1–1 (http://r-survey.r-forge.r-project.org/survey/) (S2 Additional methods). Considering the low accuracy of the Zika serological tests (due to the cross reactivity with anti-DENV antibodies) in areas of cocirculation of DENV and ZIKV, we estimated the prevalence of anti-ZIKV IgG and IgG3 through a Bayesian method for estimation of true prevalence from apparent prevalence obtained by testing individual samples [33,34]. The library prevalence (which provides frequentist and Bayesian methods useful in prevalence assessment studies) was used assuming a uniform range of sensitivity and specificity using an interval corresponding to the 95% CI of the sensitivities and specificities calculated for the Zika IgG and IgG3 tests. The beta distribution was defined based on a priori distribution of the true prevalence. According to these estimates, 10,000 first iterations were discarded and the average of the remaining 20,000, together with their respective standard deviations, were used to estimate the true prevalence of Zika by socioeconomic stratum, sex, and age group. The analysis of the convergence of estimates was performed through the Multivariate BGR statistic proposed by Gelman and Rubin [35] and improved by Brooks and Gelman [36]. We used the Pearson's Chi-square test with Rao-Scott correction [37] to compare the prevalence of Dengue, Zika and Chikungunya within and between socioeconomic strata and the *t*-Student or ANOVA tests to compare the means of the true prevalence of Zika estimated by the Bayesian method. This method proposes an adjustment of the seroprevalence estimates taking into consideration the imprecision of the calculated sensitivity and specificity. In this case, we used the 95% CI of the tests used.

The force of dengue, Zika and chikungunya infections was estimated in each socioeconomic stratum assuming a constant risk of arbovirus exposure in the study population with permanent seroconversion. The model was fitted by the effect of the sample design from a generalized linear model (GLM) with serostatus as the outcome variable (anti-ZIKV IgG and/or IgG3; anti-DENV IgG; anti-CHIKV IgM and/or IgG), a quasibinomial distribution family, complementary logit link function, and the age as exposure variable [38].

## Risk factor analysis for Zika and Chikungunya

The associations of the exposure variables with the seropositivity of Chikungunya and Zika were analyzed through hierarchical multiple regression analysis. Initially, univariate analysis was performed, calculating the crude odds ratio (OR) and the respective 95% CI for each block of variables (individual and household characteristics). The independent variables associated with the outcome at a significance level of $p < 0.25$ were included in the multivariate logistic regression model for their respective block. The variables that showed a statistically significant association with the outcome in the multiple regression models, within each block, were brought together in a new multivariate model to obtain the final model. The variables selection method used in each model regression was based in the Akaike Information Criterion (AIC).

## Results

### Characterization of the study population

A total of 2,691 residents of the selected households were eligible for the study and included 654, 1,037 and 1,000 individuals in the high, intermediate and low SES strata, respectively. Reasons for households not to participate included the difficulty in accessing the household (usually apartments in buildings) and refusals, which were more frequent in the high SES (52.4%) compared to the intermediate (18.8%) and low strata (13.6%). Among the eligible participants, those who were interviewed and consented for venous blood collection included 480 (73.3%), 815 (78.6%) and 775 (77.5%) individuals from the high, intermediate and low SES (Fig 2).

Table 1 shows the main characteristics of the studied households according to SES. We observed a gradient in the average number of residents per bedroom that increased from the high (1.36±0.03) to the low SES (1.75±0.04). Around 60% of the households in the high SES lived in high-rise apartments, while most households in the low stratum were ground level houses. Almost half of the households had no access to public sewage, with a higher proportion among those in the low stratum (62.3%). More than half of the households had irregular water supply in the low stratum (41.8%). Nearly 30% of the households were supplied by artesian well in the high socioeconomic stratum. Heads of family from the high stratum had a higher educational level and income when compared to those from the intermediate and low socioeconomic strata.

### Seroprevalence of Dengue, Zika and Chikungunya

Out of 2,070 participants, 1,837 had serological markers of previous DENV infection (anti-DENV NS1 IgG), that corresponded to an overall weighted prevalence of 88.7% (95%CI:

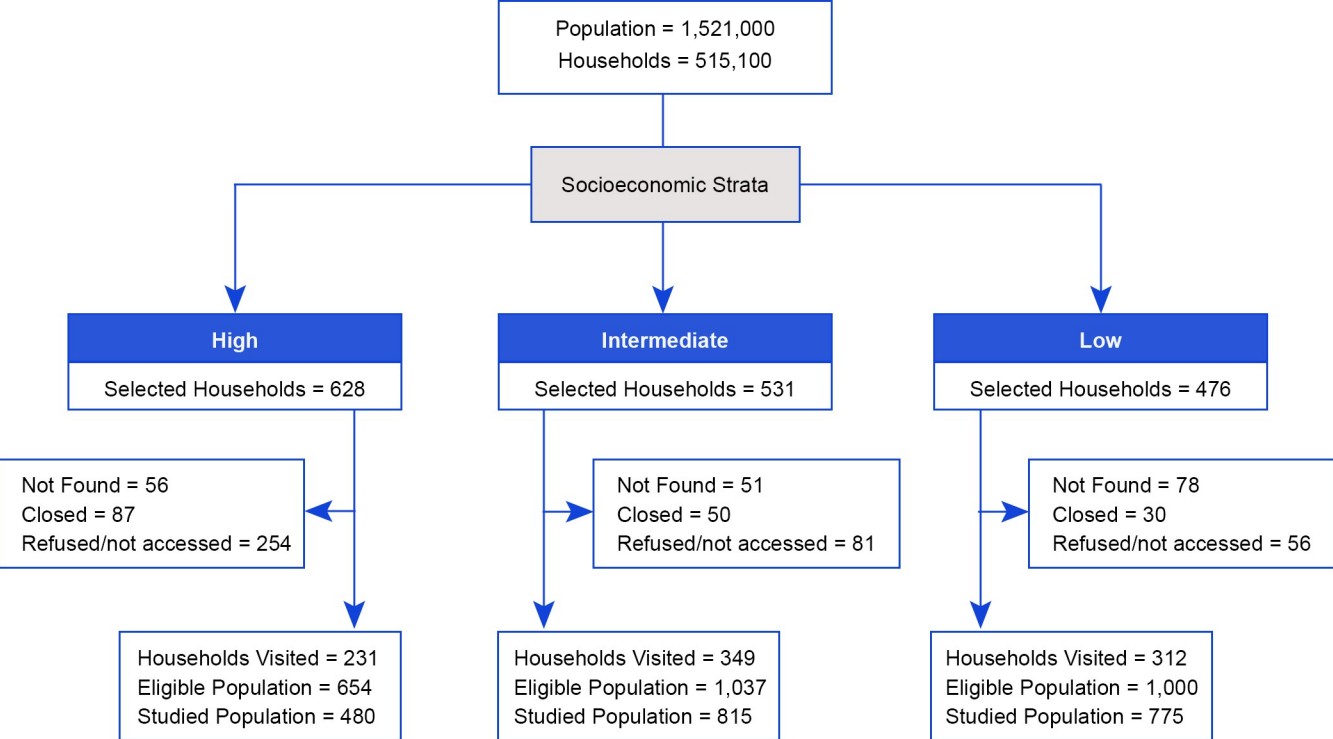

**Fig 2. Population, census tracts and calculated samples according to socioeconomic strata in Recife, Brazil, 2018–2019.**

**Table 1. Characteristics of the selected households according to socioeconomic strata in Recife, Brazil, 2018–2019.**

| Characteristics | Total | Socioeconomic Strata | | | p-value |
|---|---|---|---|---|---|
| | | High | Intermediate | Low | |
| **Studied households. n (%)** | 892 (100.0) | 231 (25.9) | 349 (39.1) | 312 (35.0) | |
| **Residents per bedroom, mean (se)** | 1.61 (0.03) | 1.36 (0.04) | 1.65 (0.04) | 1.75 (0.04) | <0.001 |
| **Type of household, n (%)** | | | | | |
| Apartment | 202 (22.8) | 146 (63.5) | 67 (19.4) | 12 (3.8) | <0.001 |
| House | 682 (77.2) | 84 (36.5) | 279 (80.6) | 300 (96.2) | |
| **Sewerage, n (%)*** | | | | | |
| Public network | 474 (56.0) | 157 (75.9) | 204 (60.7) | 116 (37.7) | |
| Other waste destinations | 373 (44.0) | 49 (24.1) | 132 (39.3) | 192 (62.3) | <0.001 |
| **Water supply, n (%)** | | | | | |
| Public network | 755 (85.0) | 157 (68.3) | 319 (92.0) | 379 (89.4) | |
| Other sources (well, others) | 133 (15.0) | 73 (31.7) | 27 (7.8) | 33 (10.6) | <0.001 |
| Regular water supply, **n (%)**** | 667 (75.2) | 157 (89.1) | 281 (81.2) | 181 (58.2) | <0.001 |
| Household garbage collection. n (%) | 832 (93.7) | 227 (98.7) | 316 (91.3) | 289 (92.6) | <0.001 |
| **Characteristics of the head of the family, n (%)** | | | | | |
| **Sex** | | | | | |
| Female | 484 (54.7) | 98 (49.4) | 178 (57.3) | 208 (55.4) | |
| Male | 401 (45.3) | 101 (50.6) | 133 (42.7) | 167 (44.6) | 0.298 |
| **Self-reported race/skin color** | | | | | |
| Mixed race (Brown) | 451 (50.6) | 92 (39.8) | n(47.0) | 195 (62.5) | |
| Black | 125 (14.0) | 20 (8.7) | 50 (14.3) | 55 (17.6) | |
| White | 283 (31.7) | 107 (46.3) | 122 (35.0) | 54 (17.3) | |
| Others/not informed/ignored | 33 (3.7) | 12 (5.2) | 13 (3.7) | 8 (2.6) | 0.001 |
| **Monthly income in minimum wages** | | | | | |
| ≤ 2 | 550 (62.4) | 57 (28.7) | 181 (58.7) | 312 (85.3) | |
| 2–4 | 185 (21.0) | 45 (22.6) | 84 (27.2) | 56 (15.1) | |
| 4–20 | 146 (16.6) | 97 (48.7) | 44 (14.2) | 6 (1.6) | 0.001 |
| **Schooling** | | | | | |
| University | 238 (27.0) | 133 (67.2) | 86 (27.6) | 19 (5.1) | |
| High school | 299 (33.9) | 36 (18.3) | 112 (35.9) | 151 (40.5) | |
| Fundamental | 345 (39.1) | 28 (14.4) | 113 (36.5) | 203 (54.3) | 0.001 |

87.0%-90.4%), consistent with the seroprevalence levels determined in 2005/2006 [12,39]. Dengue seroprevalence ranged from 81.2% (95% CI: 76.9%-85.6%) in the high SES, to 90.7% (95%CI: 88.3%-93.2%) in the low SES (Fig 3, Table 2).

A total of 1,043 participants had previous ZIKV infection (anti-ZIKV IgG), yielding an overall weighted seroprevalence of 50.4% (95%CI: 47.2%-53.6%) (Fig 3, Table 2) and a sensitivity and specificity adjusted prevalence of ZIKV infection (anti-ZIKV NS1 IgG and/or IgG3) of 38.6% (95%CI: 22.8%-54.2%) (Table 3). According to socioeconomic strata, the weighted Zika seroprevalence (IgG) was significantly lower in the high SES when compared to the intermediate and low strata (Fig 3, Table 2). The weighted seroprevalence of recent ZIKV infection (anti-ZIKV IgG3) was 5.0% (95% CI: 3.9%-6.2%); with small variations from 4.0% to 6.9% in the different age groups (Table 2). The sensitivity and specificity adjusted seroprevalence of Zika recent infection (anti ZIKV-IgG3) was 1.5% (95% CI: 0.1% -3,7%) (Table 3).

In the female population of reproductive age, the seroprevalence of Zika corrected by the design effect was 52.8% (95%CI: 48.5%-57.1%). The seroprevalence of Zika (anti-ZIKV NS1 IgG and/or IgG3) adjusted by the test accuracy was 44.8% (95%CI: 29.5%-59.2%).

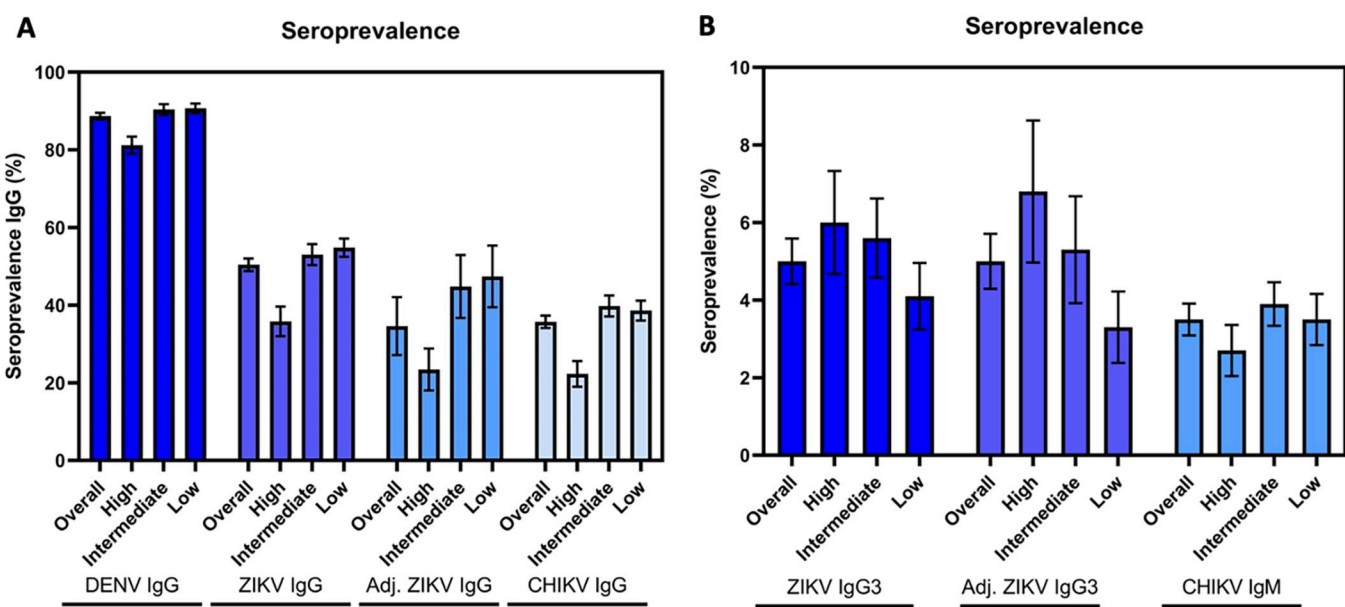

**Fig 3. Weighted Zika, Dengue and Chikungunya seroprevalence according to socioeconomic strata.** (A) Weighted IgG seroprevalence of Chikungunya Zika and Dengue including adjusted seroprevalence of Zika and (B) weighted seroprevalence of Chikungunya and Zika markers of recent infection (IgG3 and IgM) according to socioeconomic strata. Recife, Brazil, 2018–2019. Error bars indicate the standard deviations.

The weighted seroprevalence for ZIKV among children (<15 years old) was 24.6% (95% CI:18.4–30.9%), which is significantly lower than the observed prevalence for young adults (14–24 years, 45.3% (95%CI:38.8–51.9%)) (Table 2). The adjusted seroprevalence based on

**Table 2. Weighted seroprevalence of Dengue (anti-DENV IgG), Zika (Anti-ZIKV IgG e Anti-ZIKV IgG3) and Chikungunya (Anti-CHIKV IgG e Anti-CHIKV IgM) according to socioeconomic strata, sex and age group.** Recife, Brazil, 2018–2019.

| | Examined | Dengue | | Zika | | | | Chikungunya | | | |
|---|---|---|---|---|---|---|---|---|---|---|---|
| | | Anti-DENV IgG[1] | | Anti-ZIKV IgG[1] | | Anti-ZIKV IgG3[1] | | Anti-CHIKV IgG[1] | | Anti-CHIKV IgM[1] | |
| | | Pos[2] | Prev[3] (CI95%) | Pos[2] | Prev[3] (CI95%) | Pos[2] | Prev[3] (CI95%) | Pos[2] | Prev[3] (CI95%) | Pos[2] | Prev[3] (CI95%) |
| **Overall** | 2,070 | 1,837 | 88.7 (87.0–90.4) | 1,043 | 50.4 (47.2–53.6) | 104 | 5.0 (3.9–6.2) | 740 | 35.7 (32.6–38.9) | 72 | 3.5 (2.7–4.2) |
| **Socioeconomic strata** | | | | | | | | | | | |
| High | 416 | 338 | 81.2 (76.9–85.6) | 149 | 35.8 (28.3–43.3) | 25 | 6.0 (3.4–8.7) | 93 | 22.3 (15.8–28.8) | 11 | 2.7 (1.4–4.0) |
| Intermediate | 726 | 657 | 90.4 (87.7–93.2) | 385 | 53.0 (47.7–58.4) | 41 | 5.6 (3.6–7.7) | 289 | 39.8 (34.5–45.0) | 29 | 3.9 (2.8–5.1) |
| Low | 928 | 842 | 90.7 (88.3–93.2) | 509 | 54.8 (50.2–59.5) | 38 | 4.1 (2.4–5.8) | 358 | 38.6 (33.6–43.6) | 32 | 3.5 (2.2–4.8) |
| **Sex** | | | | | | | | | | | |
| Female | 1,212 | 1,099 | 90.7 (88.6–92.7) | 616 | 50.8 (47.3–54.4) | 61 | 5.1 (3.7–6.4) | 439 | 36.2 (32.8–39.7) | 51 | 4.2 (3.0–5.4) |
| Male | 858 | 738 | 86.0 (83.1–88.8) | 427 | 49.7 (45.8–53.6) | 43 | 5.0 (3.3–6.7) | 301 | 35.0 (31.0–39.0) | 21 | 2.5 (1.5–3.5) |
| **Age Group (in years)** | | | | | | | | | | | |
| 5–14 | 264 | 140 | 52.9 (45.8–60.1) | 65 | 24.6 (18.4–30.9) | 11 | 4.4 (1.7–7.0) | 80 | 30.5 (24.4–36.6) | 8 | 3.2 (1.1–5.2) |
| 15–24 | 357 | 319 | 89.3 (85.3–93.2) | 162 | 45.3 (38.8–51.9) | 14 | 4.0 (1.7–6.3) | 121 | 34.0 (28.7–39.3) | 16 | 4.5 (2.3–6.7) |
| 25–34 | 323 | 300 | 92.9 (90.1–95.8) | 160 | 49.7 (42.9–56.4) | 13 | 4.0 (1.7–6.3) | 118 | 36.6 (31.3–41.9) | 13 | 4.0 (2.2–5.9) |
| 35–44 | 375 | 360 | 96.0 (93.8–98.1) | 205 | 54.8 (49.1–60.6) | 26 | 6.9 (4.1–9.8) | 126 | 33.5 (27.9–39.1) | 14 | 3.8 (2.1–5.5) |
| 45–54 | 387 | 373 | 96.4 (94.5–98.2) | 235 | 60.6 (55.1–66.0) | 19 | 4.9 (2.8–7.0) | 148 | 38.2 (33.2–43.3) | 10 | 2.5 (0.9–4.1) |
| 55–65 | 364 | 345 | 94.8 (92.5–97.0) | 215 | 59.2 (53.1–65.3) | 21 | 5.7 (3.0–8.4) | 145 | 40.0 (33.1–46.9) | 11 | 2.9 (1.3–4.6) |

[1] All estimates were corrected for the sampling design effect.

[2] Pos stands for Positive.

[3] Prev stands for Prevalence.

**Table 3. Sensitivity and specificity adjusted seroprevalence of Zika according to socioeconomic strata, sex and age groups.** Recife, Brazil, 2018–2019.

| | Total tested | Zika | | | | | |
| --- | --- | --- | --- | --- | --- | --- | --- |
| | | IgG-ELISA[1] | | Anti-ZIKV IgG3[2] | | Anti-ZIKV IgG and/or IgG3[1] | |
| | | Pos[3] | Prev[4] (CI95%) | Pos[3] | Prev[4] (CI95%) | Pos[3] | Prev[4] (CI95%) |
| **Overall** | 2,070 | 1,043 | 34.6 (20.0–50.9) | 104 | 1.5 (0.1–3.7) | 1,073 | 38.6 (22.8–54.2) |
| **Socioeconomic strata** | | | | | | | |
| High | 416 | 149 | 23.4 (12.8–33.8) | 25 | 6.8 (3.2–11.3) | 158 | 26.3 (14.9–37.1) |
| Intermediate | 726 | 385 | 44.8 (28.9–59.1) | 41 | 5.3 (2.6–8.5) | 399 | 47.5 (31.4–61.9) |
| Low | 928 | 509 | 47.4 (31.8–61.5) | 38 | 3.3 (1.5–5.6) | 516 | 48.2 (33.1–62.6) |
| **Sex** | | | | | | | |
| Female | 1,212 | 616 | 36.5 (20.6–52.4) | 61 | 1.6 (0.1–4.0) | 633 | 38.6 (22.9–55.3) |
| Male | 858 | 427 | 33.9 (18.2–50.3) | 43 | 1.7 (0.1–4.4) | 440 | 36.6 (20.8–53.2) |
| **Age Group (in years)** | | | | | | | |
| 5–14 | 264 | 65 | 13.2 (6.3–21.7) | 11 | 6.4 (2.7–11.3) | 71 | 15.3 (7.7–24.3) |
| 15–24 | 357 | 162 | 35.5 (20.6–49.1) | 14 | 5.1 (2.2–9.2) | 168 | 37.3 (22.2–51.0) |
| 25–34 | 323 | 160 | 40.6 (24.9–55.3) | 13 | 5.6 (2.4–9.8) | 163 | 41.9 (26.0–57.2) |
| 35–44 | 375 | 205 | 47.5 (31.3–63.1) | 26 | 8.2 (4.0–13.3) | 213 | 50.6 (34.3–66.2) |
| 45–54 | 387 | 235 | 55.5 (39.1–72.2) | 19 | 5.8 (2.7–10.0) | 240 | 57.9 (41.3–74.7) |
| 55–65 | 364 | 215 | 53.8 (37.1–70.1) | 21 | 6.9 (3.3–11.6) | 220 | 55.8 (39.5–71.9) |

[1] Corrected by the effect of sensitivity assuming uniform variation between 75% and 100% and uniform specificity between 60%-80%.

[2] Corrected by the effect of sensitivity assuming uniform variation between 60% and 95% and uniform specificity between 88%-97%.

[3] Pos stands for Positive.

[4] Prev stands for Prevalence.

anti-ZIKV IgG and IgG3 also showed a remarkable increase according to age groups. The Zika seroprevalence in the age group of 5 to 14 years of age was 15.3% (95%CI:7.7–24.3), while the prevalence in the 15 to 24 years of age group was 37.3% (95%CI:22.2–51.0).

A total of 770 participants had markers of previous CHIKV infection (anti-CHIKV IgG and/or IgM), resulting in an overall weighted seroprevalence of 37.2% (95% CI: 34.0% - 40.4%). The prevalence of previous CHIKV infection (anti-CHIKV IgG) was significantly lower in the high SES when compared to the intermediate and low strata. The prevalence of recent CHIKV (anti-CHIKV IgM) infection was 3.5% (95%CI: 2.7%-4.2%), with no difference among strata (Fig 3, Table 2). The presence of markers for past and recent infection for dengue, Zika and chikungunya between females and males was found similar. Unlike to what was observed with DENV and ZIKV in low and intermediate SES, the CHIKV seroprevalence significantly increased with age (Tables 2 and 3).

## Force of infection of dengue, Zika and chikungunya

The Fig 4 and Table 4 show the age-related force of DENV, ZIKV and CHIKV infections estimated through generalized linear models (GLM) according to SES and using the serostatus (recent and/or previous infection) as the outcome variable. The seroprevalence of dengue, Zika and chikungunya showed non-linear associations with age in the three SES. Differently from what was observed in the seroprevalence of dengue and Zika, there was no significant association of chikungunya seroprevalence with age.

The seroprevalence of dengue, which started circulating in the city 4 decades before, ranged from 52.9% in the population of 5 years of age and reached 94.8% in the population of 55 to 65 years of age. These levels have been relatively stable for more than 10 years. We also observed

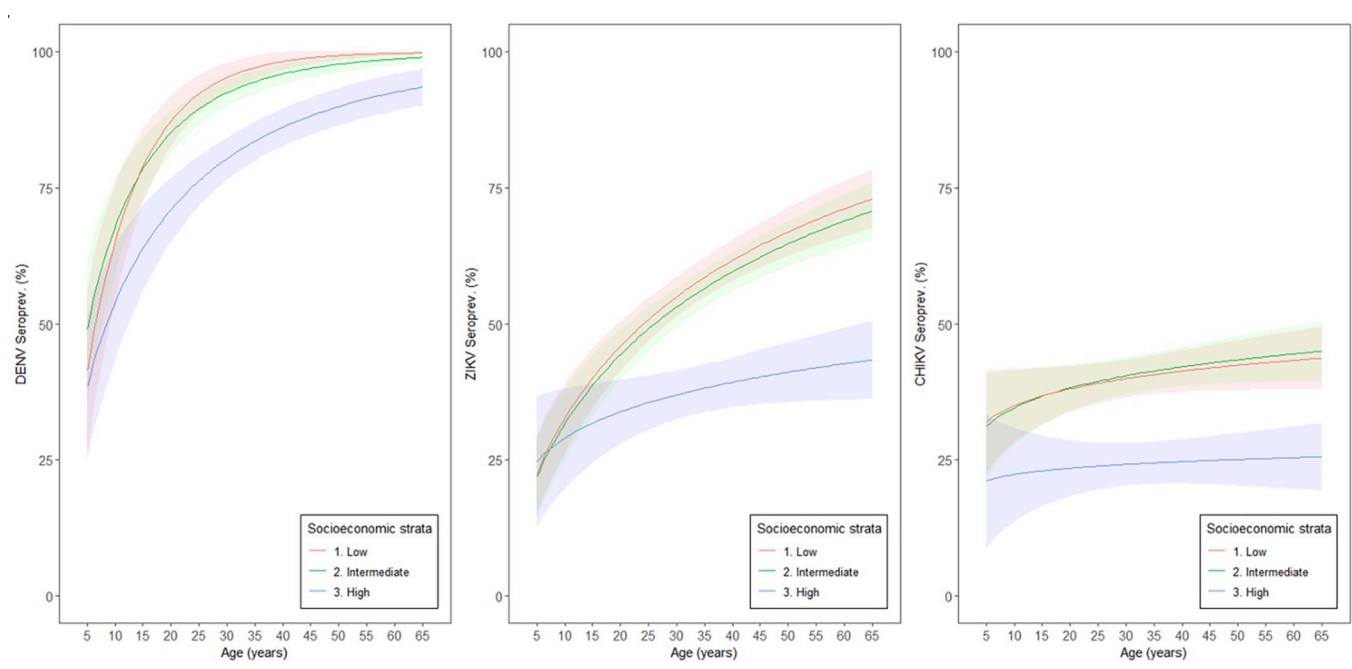

**Fig 4. Force of infection of dengue, Zika and chikungunya according to socioeconomic strata.** Recife, Brazil, 2018–2019.

that DENV seroprevalence plateau around 35 years of age in the intermediate and low SES. The estimated force of infection was estimated to be 1.4 times higher in the low SES compared to the high SES.

Similarly, the estimated force of ZIKV infection was 2.4 times greater in the low socioeconomic strata when compared to the high socioeconomic strata. The seroprevalence of infection in the high SES was below 25% at 5 years of age and did not even reach 40% at 65 years of age. The ZIKV seroprevalence in intermediate and low SES also starts below 25% at 5 years of age, but reached levels above 70% at 65 years of age. This data suggests that high Zika herd immunity in adults living in the low SES might be the main factor contributing to the cessation of the ZIKV epidemic, rather than the immunity in the whole population.

The seroprevalence of chikungunya was 21.6% at 5 years of age and slightly increased to 26.8% at 65 years of age in the high SES. In the intermediate and low SES, the seroprevalence was around 35% at 5 years, and reached approximately 45% at 65 years of age in both strata (Fig 4).

**Table 4. Force of dengue, Zika and chikungunya infections (fitted by the effect of the sample design) through a generalized linear model (GLAM).**

|  | High | | Intermediate | | Low | |
|---|---|---|---|---|---|---|
|  | **Estimate** | **p-value** | **Estimate** | **p-value** | **Estimate** | **p-value** |
| **Dengue** |  |  |  |  |  |  |
| Intercept | -1.8102 | <0.001 | -1.6031 | <0.001 | -2.1936 | <0.001 |
| Log(age) | 0.6753 | <0.001 | 0.7506 | <0.001 | 0.9736 | <0.001 |
| **Zika** |  |  |  |  |  |  |
| Intercept | -1.6993 | 0.004 | -2.4131 | <0.001 | -2.4040 | <0.001 |
| Log(age) | 0.2720 | 0.078 | 0.6274 | <0.001 | 0.6402 | <0.001 |
| **Chikungunya** |  |  |  |  |  |  |
| Intercept | -1.5682 | 0.003 | -1.2781 | <0.001 | -1.2011 | 0.003 |
| Log(age) | 0.0835 | 0.508 | 0.1828 | 0.079 | 0.1551 | 0.133 |

### Associated risk factors for ZIKV and CHIKV infections

The S2–S7 Tables show the results of the crude and adjusted analysis of the association of individual and household characteristics with ZIKV and CHIKV infection in the distinct SES.

Regarding ZIKV infection, older age and living in a house were independent risk factors for infection in the three socioeconomic strata. A dose-response gradient of the risk of exposure to ZIKV with increasing age was observed in the intermediate and low SES. Living in a house represented a 2-fold increased risk of infection in the three SES. Presence of anti-dengue antibodies represented a 3-fold increased risk for ZIKV infection among participants from the high socioeconomic stratum. Low educational level was a risk factor for infection at the individual level in the high and intermediate strata, although with weak strength of association. Higher individual and head of household income were a protective factor against ZIKV infection in the intermediate SES (S4 Table).

The S5–S7 Tables show the results of crude and adjusted analyzes of the association of household and individual factors with CHIKV infection. The risk of CHIKV infection was approximately three times higher among those living in a house compared to those individuals living in apartments in the high (adjusted OR 2.64; 95%CI: 1.34–5.17) and intermediate (adjusted OR 3.23; 95% CI: 1.80–5.50) SES. Most of the participants in the low SES lived in households (96.8%), and no CHIKV positive was detected among those who lived in an apartments in this area.

Considering the household-related characteristics associated with CHIKV infection, lack of access to the public sewage system and the lower level of schooling of the head of the household were risk factors for infection in the high SES. There was a negative association between the head of the family's income and risk of CHIKV infection in the low stratum (S7 Table).

In the intermediate SES, the presence of the serological marker of DENV infection represented a risk of CHIKV infection twice as high when compared to those without DENV infection (adjusted OR 1.97; 95%CI: 1.28–3.02).

In the high SES, participants with fundamental/illiterate level of education had higher risk of CHIKV infection when compared to those with university level. The report of daily use of repellent was a protective factor in comparison to those who did not use it (adjusted OR 0.22; 95%CI: 0.05–0.87). This use of repellent was not associated with infection in the other strata (S7 Table).

## Discussion

This population-based survey estimated dengue, Zika, and chikungunya seroprevalence in a hyperendemic urban area of dengue and the epicenter of the Zika-related microcephaly epidemic in Brazil. The study, conducted after nearly four decades of DENV circulation and two years after the introduction of ZIKV and CHIKV, confirmed the intense transmission of arboviruses in this setting. After the first epidemic wave of ZIKV and CHIKV, 34.6% (adjusted anti-ZIKV IgG) and 35.7% (anti-CHIKV IgG) of the population residing in the city show serological markers of previous infection by these viruses, demonstrating the high vulnerability of this population to diseases transmitted by *Aedes aegypti*. The study also provided evidence of the persistence of low-level co-circulation of ZIKV and CHIKV, measured by serological markers of recent infection (anti-ZIKV IgG3 and anti-CHIKV IgM), during an interepidemic period, consistent with the numbers of notified cases reported by the official surveillance system in this setting [40].

Our study has some limitations inherent of serosurveys in large urban settings. Firstly, children below five years of age were not eligible due to the difficulty of obtaining blood specimens during household visits. As expected in large urban areas, some dwellings could not be

accessed, which is partially due to a relatively high rate of refusals to participate in the high socioeconomic stratum compared to other strata. Nevertheless, the data presented is representative of the general population. The Zika serologic tests were optimized and calibrated using a validated panel of well characterized dengue and Zika samples from the same area and the results were correlated to PRNT data. We further adjusted the prevalence of Zika for a Bayesian model to account for the sensitivity and specificity of the ELISA test.

In a previous population-based survey in the city of Recife in 2006, around 15 years prior this one, we estimated that almost the entire population had DENV IgG antibodies for one or more DENV serotypes [12,28]. A small hospital-based study in low SES pregnant women found similar levels of almost 100% of DENV exposure in 2010 in this population [41]. This level of seroprevalence could be classified as one of the highest DENV seroprevalence rates worldwide and only comparable to other highly endemic countries in America, such as Trinidad, French Caribbean, Ecuador, and Suriname [42,43].

In this population-based survey conducted in 2018–2019, the overall ZIKV seroprevalence (weighted 50.4%) suggests an intense virus exposure during the first wave of ZIKV epidemic in 2015–2016 in Recife, Northeast Brazil. This prevalence remains high even when adjusting for the non-optimal accuracy of the performed serological test (adjusted 34.6%). We highlight that the adjusted seroprevalence for women at reproductive age reached close to 45% of ZIKV infection in this population. This high level of ZIKV exposure may explain one of the highest prevalence of adverse pregnancy outcomes, such as microcephalic cases, reported during the first epidemic wave in Brazil [15]. The results of different studies may not be comparable due to the distinct laboratory tests applied and/or study designs. In our setting, a case-control study conducted in 2016 mainly concentrated in low SES participants, found 57.2% of seroprevalence of ZIKV virus by PRNT among pregnant women with non-adverse outcomes (control sample) during the peak of the epidemic [44]. Netto and al [45] reported high ZIKV seroprevalence (63%) using ELISA and PRNT tests among a convenience sample in Salvador, Bahia, estimating a reproduction number of 2.1 during the outbreak.

The levels of ZIKV seroprevalence found in our study are in line with results reported in the adult population in the large urban center of Nicaragua after the 2016 epidemic wave. Other international post-epidemic surveys pointed out to higher seroprevalence levels such as the results reported after outbreaks in Yap Island [46], Pacific Region and French Polynesia, Micronesia [47]. Interestingly, we found at least 1.5% (adjusted) prevalence of a recent marker of infection (IgG3) suggesting the persistence of ZIKV circulation in our setting after the 2015/2016 epidemic. This finding is corroborated by the report of the Pan American Health Organization of almost 30,000 cases in 2022. The health secretary of the state of Pernambuco was notified of 4,570 cases in 2022, with most of them in Recife. The incidence of ZIKV viremia using molecular tests was undetectable among health blood donors in the same setting and period of our study [48]. The likely explanation may be the short duration of viremia in a self-selected healthy population as blood donors. However, the serological results support that ZIKV has been circulating at lower levels even after a large epidemic. ZIKV circulation after a large outbreak has been a matter of discussion in the recent literature [49] and deserves further population-based studies.

In our densely urbanized setting with high infestation of *Aedes aegypti*, the CHIKV displaced ZIKV outbreak (2015–2016) [5]. The current survey showed high prevalence of CHIKV infection (35.7%), with little variation among age groups as expected considering a recently introduced virus. This level of infection was not sufficient to avoid the occurrence of another CHIKV epidemic detected (2021) by the official surveillance, approximately one year after our survey [50]. In a meta-analysis of CHIKV seroprevalence studies conducted in Brazil, the estimated overall prevalence was 24% including three previous studies [43].

Another interesting finding was 3.5% prevalence of recent CHIKV infection and the simultaneous low-level co-circulation of CHIKV and ZIKV in our setting. Concurrently to this survey, we also described the co-circulation of ZIKV and CHIKV among pregnant women in a maternity-based study conducted in this same city [51]. In fact, the circulation of CHIKV during ZIKV (2015–2017) outbreak was previously documented among pregnant women with rash notified by the official surveillance system in our setting [52].

The introduction of CHIKV in Recife is quite recent compared to other regions such as Southeast Asia where this virus has been circulating for several decades [53]. Our study showed evidence that one third of the population, i.e., around 544 thousand people living in the city had been exposed to CHIKV since the first epidemic wave (2016) [54]. This figure is 45-fold higher than the 11,984 accumulated reported CHIKV suspected cases reported by the surveillance system since 2015 [50], until the end of the current survey (Feb. 2019). This estimated ratio between CHIKV infection and notified cases suggests substantial case underreporting and/or high frequency of subclinical/inapparent infections. Indeed, asymptomatic infections can represent a very large fraction of CHIKV cases. A cohort study in Managua, Nicaragua identified 49% of asymptomatic CHIKV infections, and this high level of asymptomatic cases may explain the high level of sub-notification. Moreover, they also demonstrated that the proportion of asymptomatic infections depend on the CHIKV lineage [55]. We have only started investigating the genotype of the CHIKV strains circulating in the region [56].

We did not find statistical differences between the sexes regarding the seroprevalence of the three arboviruses surveyed. This result is in accordance with previous population surveys conducted in this setting [12], in other Brazilian states [10,57] and other countries in the region [58]. Conversely, population-based studies conducted in the Americas reported higher seroprevalence of ZIKV and CHIKV in males compared to females [59,60]. This difference can be explained by local characteristics of the population or by methodological differences across the studies.

Interestingly, our analysis showed different age-related infection curves for dengue, Zika or chikungunya. In the group of 5 to 14 years of age, 52.9% of the children had markers of previous DENV infection, while more than 90% of the population with 25 to 65 years of age had been infected. These findings are in consonance with the intense circulation of the virus in this population for four decades. A similar age-related pattern for dengue infection was also documented by our research team in this setting in 2006 [12,29]. In addition, we observed a steady increase in the force of ZIKV infection from less than 25% of prevalence at the age of 5 years to approximately 70% at the age of 65, confirmed by the results of the regression model (Tables 4 and S4). This finding was consistent with the results of the multiple regression analysis, which, unlike that observed for CHIKV infection, showed a higher risk of infection in the population aged 15 years and older when compared to the population below this age range. Although this finding seems unexpected for a recent introduced virus, it is in line with results from Nicaragua [59] and Puerto Rico [60]. The significant increase in the seroprevalence between children and young adults may suggest that anti-dengue antibodies or sexual transmission may have influenced the transmission of Zika. Seroprevalence levels around 60–70% which suggest herd immunity are only found in (older) adults of low and intermediate SES, suggesting this group as the major driver for population immunity. In addition, higher levels of anti-dengue antibodies can provide partial protection against ZIKV also contributing to the overall herd immunity. Other studies have reported a significant role of sexual transmission of Zika [61,62], which could also contribute to the higher seroprevalence in (older) adults. Another population survey conducted in French Guiana did not show evidence of increased seroprevalence with age [58], and it still controversial if sexual transmission plays a significant role for ZIKV

epidemiology [63,64]. In contrast, the CHIKV age-related curve of infection is almost as a straight line, which is compatible with the recent introduction of the virus in this region.

The weighted seroprevalence of dengue, Zika and chikungunya was higher in areas classified as low and intermediate socioeconomic levels when compared to the high socioeconomic strata. The latter has greater coverage of sanitation, regular water supply and most of the population live in apartment buildings. In general, our data also reinforce the role of unplanned urbanization and poverty as one of the factors that influence the incidence and expansion of arboviruses [65,66].

The analysis of the association between individual and household characteristics and DENV, ZIKV and CHIKV infections showed different patterns. ZIKV and CHIKV infections were associated with lower educational levels as an indicator of health inequities and an independent risk factor for infection in almost all socioeconomic strata. In addition, living in a low-rise household instead of a high-rise apartment yielded a three-fold increased risk of exposure to CHIKV or to ZIKV infection. We also found that living in a house compared to high rise flat was a risk factor for DENV infection in the previous survey (2005/2006) in the city of Recife [12]. These findings are in line with results from other surveys in localities in southeastern Brazil [67,68] and in the city of Singapore in Asia [69,70]. In our setting, the greater risk of exposure to these arbovirus infections among residents of houses (one-story dwellings) can be explained by the fact that, unlike the areas of the high stratum, most of the households located in the intermediate and low socioeconomic strata consist of this type of residence (more than 80%). These densely populated areas, with less sanitation coverage and irregular water supply, provide the most favorable environmental conditions for the proliferation of *Aedes* breeding sites [65]. Another possible explanation would be the greater proximity to Aedes breeding sites in a household compared to high rise flats located above the ground floor. The identification of the risk factors associated with arboviral infections allow a more effective and precise use of control measures. Indeed, Bisanzio et al using spatio-temporal analyzes demonstrated high heterogeneity in arboviral transmission in Merida, Mexico. Using geocoded arboviral notification data over time, they identified areas with high transmission and others with low transmission in the same urban setting [71]

In conclusion, our household-survey highlights the high vulnerability of these urban population to *Aedes*-borne arbovirus infections in the Northeast region of Brazil. Also, our results provide evidence of the persistence of the co-circulation of ZIKV and CHIKV in this highly urbanized setting, three years after the peak of ZIKV epidemic. Considering the large proportion of susceptible population for ZIKV and CHIKV infection, it should be an alert for future outbreaks. There is an urgent need of new approaches to *Aedes* surveillance and control, besides the development and efficacy assessment of vaccines against these arboviruses. The planning and implementation of intersectoral and integrated interventions for the control of diseases transmitted by *Aedes* in poor urban settings should be crucial to reduce the burden of arbovirus disease and mortality in endemic countries.

## Supporting information

**S1 Fig. ROC curve analysis of the Euroimmun ZIKV IgG (A), ZIKV NS1 IgG3 (B) and DENV 1–4 NS1 total IgG ELISA assays.** The paired results for sensitivity and specificity were plotted as points in a ROC space and the trade-off between these measures for different discrimination cut-offs are graphically represented. The red dotted line represents the cut-off value.
(TIF)

**S1 Table. List of exposure variables.**
(DOCX)

**S2 Table. Crude analysis of the association between household characteristics and ZIKV infection.** Recife, Brazil, 2018–2019.
(DOCX)

**S3 Table. Crude analysis of the association between individual characteristics and ZIKV infection.** Recife, Brazil, 2018–2019.
(DOCX)

**S4 Table. Final model of the association of household and individual characteristics with ZIKV infection.** Recife, Brazil, 2018–2019.
(DOCX)

**S5 Table. Crude analysis of the association between household characteristics and CHIKV infection.** Recife, Brazil, 2018–2019.
(DOCX)

**S6 Table. Crude analysis of the association between individual characteristics and CHIKV infection.** Recife, Brazil, 2018–2019.
(DOCX)

**S7 Table. Final model of the association of household and individual characteristics with CHIKV infection.** Recife, Brazil, 2018–2019.
(DOCX)

**S1 Additional methods. ZIKV NS1 IgG3 assay, DENV NS1 total IgG assay and Plaque reduction neutralization test (PRNT).**
(DOCX)

**S2 Additional methods. Packages for Prevalence Data Analysis using R Programme.**
(DOCX)

## Acknowledgments

We thank Andre Sá de Oliveira for generating the Fig 1. We also thank Dylan Tuttle and Dr. Priscila M. Castanha for reviewing the text and generating Figs 2 and 3.

## Author Contributions

**Conceptualization:** Cynthia Braga, Celina M. T. Martelli, Wayner V. Souza, Maria de Fatima P. M. Albuquerque, Carlos Frederico C. A. Melo, Thomas Jaenisch, Ernesto T. A. Marques.

**Data curation:** Cynthia Braga, Celina M. T. Martelli, Wayner V. Souza, Carolline A. Mariz, Isabelle F. T. Viana.

**Formal analysis:** Cynthia Braga, Wayner V. Souza, Carlos F. Luna, Isabelle F. T. Viana.

**Funding acquisition:** Cynthia Braga, Roberto D. Lins, Jan Felix Drexler, Thomas Jaenisch, Ernesto T. A. Marques.

**Investigation:** Cynthia Braga, Carolline A. Mariz, Clarice N. L. Morais, Ernesto T. A. Marques, Isabelle F. T. Viana.

**Methodology:** Cynthia Braga, Wayner V. Souza, Isabelle F. T. Viana.

**Project administration:** Cynthia Braga, Roberto D. Lins, Thomas Jaenisch, Isabelle F. T. Viana.

**Supervision:** Cynthia Braga, Carolline A. Mariz, Clarice N. L. Morais, Isabelle F. T. Viana.

**Validation:** Isabelle F. T. Viana.

**Writing – original draft:** Cynthia Braga, Celina M. T. Martelli, Wayner V. Souza, Carlos F. Luna, Carolline A. Mariz, Roberto D. Lins, Ernesto T. A. Marques, Isabelle F. T. Viana.

**Writing – review & editing:** Cynthia Braga, Celina M. T. Martelli, Wayner V. Souza, Carlos F. Luna, Maria de Fatima P. M. Albuquerque, Carolline A. Mariz, Clarice N. L. Morais, Carlos A. A. Brito, Carlos Frederico C. A. Melo, Roberto D. Lins, Jan Felix Drexler, Thomas Jaenisch, Ernesto T. A. Marques, Isabelle F. T. Viana.

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
