## [Decision Letter · Decision Letter 0]

25 Apr 2023

Dear Dr Marques,

Thank you very much for submitting your manuscript "Seroprevalence of Dengue, Chikungunya and Zika at the epicenter of the congenital microcephaly epidemic in Northeast Brazil: a population-based survey" for consideration at PLOS Neglected Tropical Diseases. As with all papers reviewed by the journal, your manuscript was reviewed by members of the editorial board and by several independent reviewers. In light of the reviews (below this email), we would like to invite the resubmission of a significantly-revised version that takes into account the reviewers' comments. 

We cannot make any decision about publication until we have seen the revised manuscript and your response to the reviewers' comments. Your revised manuscript is also likely to be sent to reviewers for further evaluation.

Sincerely,

Renata Rosito Tonelli, PhD

Academic Editor

Elvina Viennet

Section Editor

Dear Dr Marques,

As you will see from their reports, the reviewers’ recommendations are mixed. While referees #2 and #3 enthusiastically support publication, referee #1 considers that your study is flawed and rejected its publication in PNTD. After carefully reading your study, I agree with reviewers #2 and #3 who recommend that a revised version of the manuscript can be reconsidered. If you think that you can deal satisfactorily with the criticisms on revision, I would be pleased to see a revised manuscript. For further clarification, I ask that you respond to the following comment of reviewer #1: "In a dengue endemic population where all four dengue viruses have been transmitted, a survey of antibodies will detect many, many different patterns of prior infections. Distinguishing these patterns and attributing them to primary or secondary (dengue then Zika or Zika then dengue) will be very difficult and will require much more foundational data than is presented here"

Reviewer's Responses to Questions

**Key Review Criteria Required for Acceptance?**

**Methods**

-Are the objectives of the study clearly articulated with a clear testable hypothesis stated?

-Is the study design appropriate to address the stated objectives?

-Is the population clearly described and appropriate for the hypothesis being tested?

-Is the sample size sufficient to ensure adequate power to address the hypothesis being tested?

-Were correct statistical analysis used to support conclusions?

-Are there concerns about ethical or regulatory requirements being met?

Reviewer #1: In a dengue endemic population where all four dengue viruses have been transmitted, a survey of antibodies will detect many, many different patterns of prior infections. Distinguishing these patterns and attributing them to primary or secondary (dengue then Zika or Zika then dengue) will be very difficult and will require much more foundational data than is presented here. The serological methods used do not describe the attributes of primary or secondary dengue infections or show how they can be identified nor does it describe how dengue antibodies can be distinguished from antibody responses to Zika virus infections in individuals who either were seronegative or immune to one or more dengue viruses when infected. This cannot be done, as alleged, by use of algorithms. What is required are robust and reproducible serological methods.

Reviewer #2: This article is the result of a very interesting and breakthrough study examining the seroprevalence of Dengue, Chikungunya and Zika in the Northeast of Brazil, a very poor region, at the epicenter of the congenital microcephaly epidemic. The study required the design of an appropriate methodological strategy to describe and discuss the consequences of simultaneous exposure and vulnerability to the three arboviral infections in the same population, in different age groups and socio-economic strata. The methodological strategy is clear and precise, supported by a well designed population-based survey. The objectives are clearly presented and articulated with the testable hypotheses stated. The population is described in a clear and precise way, appropriate for the hypotheses being tested. The statistical analysis used to support conclusions is correct and well conceived. There are no concerns about ethical or regulatory requirements.

Reviewer #3: The manuscript Seroprevalence of Dengue, Chikungunya and Zika at the epicenter of the congenital microcephaly epidemic in Northeast Brazil: a population-based survey shows relevant aspects of the transmission dynamic from a hyperendemic arbovirus from Brazil, where the main urban arbovirus, dengue, Zika, and Chikungunya, has coincided. 

As part of the methodology, the authors chose a seroprevalence study to identify the principal risk factors associated with arbovirus infection by adopting a multistage random stratification in two steps. First selecting, the units of study ( named census tract CT), and in the second stage, the households that included all residents' target age of the study. 

With this methodology, the authors have gotten a sample size that allowed to address with enough power the hypothesis of the study. 

Also, the authors observed all ethical aspects of the research protocol.

**Results**

-Does the analysis presented match the analysis plan?

-Are the results clearly and completely presented?

-Are the figures (Tables, Images) of sufficient quality for clarity?

Reviewer #1: This is a flawed attempt to describe recent dengue 1 – 4, chikungunya and Zika virus epidemiology in residents of Recife, Brazil, by measuring prevalence of antibodies generated by endemic DENVs and virgin-soil outbreaks of chikungunya and Zika viruses. How to distinguish dengue from Zika antibodies and vice versa in humans is a well-known unsolved serological problem. This is the crucial challenge of this manuscript. It is a challenge the authors have failed. Dengue viruses circulating in Recife for several decades have produced the age-specific serological data displayed in Figure 4 A. The similar seroprevalence of age specific antibodies shown in Figures 4A and 4B are remarkable. Both describe age specific antibody distribution characteristic of virus endemicity. The increased prevalence of “Zika antibodies” in older aged individuals shown in Figure 4 B can only be explained by Zika antigenic cross-reactions with antibodies in sera likely resulting from two or more dengue infections. The Zika age-specific antibody prevalence curve should look like the antibody prevalence curve in Figure 3 C, antibodies following a single introduction into the population of chikungunya virus. Despite possible low level Zika virus infection in 2018 – 19, there is no evidence of vigorous Zika endemicity after 2016. The Zika epidemiology described in Brazil, therefore, is a classical single source epidemic.

Specific comments:

Line 115 -120. The authors cite erroneous dates of virus introduction. Data provided by PAHO document chikungunya virus introduction into the Americas and widespread infection beginning in 2014. CHIK infections continued at a high rate through 2016 and at a low rate until the present time. Zika virus was introduced in 2015, spreading with high incidence in 2016 but sharply declining in 2017. The authors describe this sequence exactly in reverse. The confusion between dates of entry of Zika and chikungunya viruses into Brazil confounds this study and requires repair. 

Line 127. “Susceptibility”

Line 229. “indetermined” – undetermined.

Reviewer #2: The results of this population-based survey are relevant and an important contribution to critical and still open questions in the literature, unveiling the dramatic consequences of simultaneous exposure of the study population to the three arboviral diseases. These results raise intriguing issues highlighted by the authors, related to the different profiles of the epidemics and the differences in exposure and vulnerability among age groups, sex and socio-economic strata. These results confirmed continued DENV transmission and intense ZIKV and CHIKV transmission during the 2015/2016 epidemics followed by ongoing low-level transmission. Issues related to herd immunity and antibody detection thresholds were examined and the results indicate a concerning scenario: a significant proportion of the population is likely still susceptible to be infected by ZIKV and CHIKV. Other intriguing issues stressed by the authors refer to the reasons underlying cease of the ZIKV epidemic in 2017/18, still an open question.The analysis is well presented and clearly matches the analysis plan. The results are clearly and completely presented. The Figures and Tables are clear and well designed.

Reviewer #3: Yes. The results were coherent with the analysis plan, and the reader can easily understand the contents in the tables and figures.

**Conclusions**

-Are the conclusions supported by the data presented?

-Are the limitations of analysis clearly described?

-Do the authors discuss how these data can be helpful to advance our understanding of the topic under study?

-Is public health relevance addressed?

Reviewer #1: This attempt to describe the serological status of the population of Recife, Brazil is not accurate as it is based upon flawed methods.

Reviewer #2: In their conclusion, the authors underline the high vulnerability of the study population to these arboviral infections in the Northeast of Brazil and provide evidence of the persistence of the co-circulation of ZIKV and CHIKV in this highly urbanized setting, three years after the peak of ZIKV epidemic, alerting to the risk of future outbreaks. Considering the large proportion of susceptible population for ZIKV and CHIKV, they stress the urgent need for Aedes surveillance and control and development and efficacy trials of vaccines against these arboviruses. The conclusions are well elaborated and supported by the data presented. The authors discuss and clearly explain how the data presented and the results of the study can be helpful to advance our understanding of the topic. Limitations of the analysis are clearly described, indicating the need for further research in this area.

Reviewer #3: The data presented confirmed a high level of exposition of the population against the main urban flavivirus infection. It further shows the level of heterogeneity of the transmission in the city's different economic and social strata.

Bisanzio D et al, 2018 ( Spatio-temporal coherence of dengue, chikungunya and Zika outbreaks in Merida, Mexico. PLoS Negl Trop Dis 12(3): e0006298. https://doi. org/10.1371/journal.pntd.0006298) also demonstrated this heterogeneity in the arbovirus dynamics transmission.

Applying different methodologies, the authors also identified areas with additional risk for dengue infection named hotspots. In a longitudinal cohort study the status for DENV infection in 505 children was compared between those residing inside and outside the hotspot areas. Living inside the hotspot area was associated with a significantly higher infection probability than living outside (odds ratio, 1.71 [95%CI, 1.08–2.20]; p < 0.05).

Similarly, in Recife ZIKV infection, the estimated force of infection was 2.4 times 449 greater in the low socioeconomic strata when compared to the high strata.

This aspect is highly relevant because it can guide local strategies in terms of policies for the prevention and control of these diseases. Moreover, focusing efforts in high transmission areas has the additional benefit of increased efficiency by better allocating limited personnel and financial resources.

Regarding chikungunya, the authors might explore further aspects of the inapparent infection. 

Bustos Carrillo F et al . 2019 ( Epidemiological evidence for lineage-specific differences in the risk of inapparent chikungunya virus infection. J Virol 93:e01622-18. https://doi.org/10.1128/JVI .01622-18) provide epidemiological evidence that chikungunya epidemics are strongly influenced by CHIKV lineage, particularly with respect to the proportion of inapparent CHIKV infections at the population level. 

I suggest that the authors identify in the article the predominant CHICKV strain during the epidemics in Recife and compare the results of inapparent infections with the literature.

**Editorial and Data Presentation Modifications?**

Reviewer #1: needs more science.

Reviewer #2: (No Response)

Reviewer #3: No suggestions

**Summary and General Comments**

Reviewer #1: This is a flawed study.

Reviewer #2: (No Response)

Reviewer #3: No comments

PLOS authors have the option to publish the peer review history of their article (what does this mean?). If published, this will include your full peer review and any attached files.

Reviewer #1: No

Reviewer #2: Yes: Cristina Possas

Reviewer #3: No
---

## [Editor Report · Decision Letter 1]

3 Jun 2023

Dear Dr Marques,

We are pleased to inform you that your manuscript 'Seroprevalence of Dengue, Chikungunya and Zika at the epicenter of the congenital microcephaly epidemic in Northeast Brazil: a population-based survey' has been provisionally accepted for publication in PLOS Neglected Tropical Diseases.

Best regards,

Renata Rosito Tonelli, PhD

Academic Editor

Elvina Viennet

Section Editor

---

## [Editor Report · Acceptance letter]

27 Jun 2023

Dear Dr. Marques,

We are delighted to inform you that your manuscript, "Seroprevalence of Dengue, Chikungunya and Zika at the epicenter of the congenital microcephaly epidemic in Northeast Brazil: a population-based survey," has been formally accepted for publication in PLOS Neglected Tropical Diseases.

Best regards,

Shaden Kamhawi

co-Editor-in-Chief

Paul Brindley

co-Editor-in-Chief
